# Regulation of Digital Healthcare in India: Ethical and Legal Challenges

**DOI:** 10.3390/healthcare11060911

**Published:** 2023-03-21

**Authors:** Dipika Jain

**Affiliations:** Jindal Global Law School, O.P. Jindal Global University, Sonepat 131001, India; djain@jgu.edu.in

**Keywords:** digital health, law, regulation, informed consent, privacy

## Abstract

In the wake of the COVID-19 pandemic, digital healthcare has gained an influx of interest and global investment. The WHO has published guidelines and recommendations for countries to successfully implement telemedicine on a large, nationwide scale. This is not only helpful for patients who wish to protect themselves from COVID-19 and related illnesses that they may be vulnerable to, but it also has great potential to increase access to healthcare. In India, a country without universal healthcare grappling with a high level of distrust in the public health system, there are several implementation challenges for digital healthcare across the country. The current laws in India that regulate technology do not explicitly address telehealth, nor are there adequate data protection laws in place that could manage the significant amount of data that would be generated by digital healthcare if applied on a large scale. Further, there are concerns at the level of patient privacy, which could be compromised through digital healthcare. In addition to the legal concerns surrounding privacy, there is no framework in place to ensure informed consent in a digital healthcare context. In this paper, I analyze the legal, structural, and ethical concerns around digital health and provide an understanding of the problems these shortcomings pose, as well as policy recommendations for overcoming these problems.

## 1. Introduction

Devarajeevanahalli, one of the largest urban slums situated in Bangalore, is characterized by poverty, overcrowding, and hazardous living conditions [1]. The health of its residents is impacted by these conditions, which also makes it difficult for healthcare workers to conduct health screenings of the community [1]. Additionally, most residents of Devarajeevanahalli do not have health insurance and therefore, frequently do not seek necessary medical care. A community-based cross-sectional survey was conducted by BMC Health to determine the prevalence of health conditions using THULSI (Toolkit for Healthy Urban Life in Slums Initiative), a mobile screening toolkit [1]. The survey reported poor income security and a huge burden of health issues. Due to the poor penetration by healthcare providers into the region, most people were unaware of their disease conditions prior to the screening [1].

The relatively simple technological solution of mobile screenings enabled local healthcare providers to screen the residents despite many pre-existing structural challenges. Household mobile health screenings may prevent chronic illness and treat diseases in their early stages, particularly in communities where access to adequate healthcare is low. This is just one example of the ways digital healthcare can increase access to necessary medical care and improve community health; ensuring the right to healthcare is paramount when developing digital models of healthcare.

Internationally, several policies and guidelines exist on digital health, with the World Health Organization (WHO) releasing a Global Strategy on Digital Health 2020–2025 [2] to strengthen health systems through the application of digital health technologies for consumers, health professionals, healthcare providers, and shifting the industry towards empowering patients and achieving the vision of health for all. It is designed for adoption even by Member States who have limited access to digital technologies, goods, and services [2]. These policies have been implemented by several countries across the globe. For instance, the laws on privacy in the EU that are applicable to EU Member States have been updated and strengthened through the adoption of the General Data Protection Regulation (GDPR), which came into force in May 2018. Digital healthcare is on the rise across Asia, especially Southeast and East Asia. In Singapore, the National Electronic Health Record (NEHR) system was rolled out in 2011, and the country has begun transitioning to private clouds to store health data. Other countries such as South Korea and Thailand also have robust models for digital health. However, in the case of India, experts have expressed concern about widespread digitization policies alluding to the significant shortages in capacity, challenges in transportation in rural areas, lack of financial resources, and stigmatization around certain conditions, among others, as possible hurdles to realizing the true potential of digital health in the country (For instance, the laws on privacy in the EU that are applicable to EU Member States have been updated and strengthened through the adoption of the General Data Protection Regulation (GDPR), which came into force in May 2018. Digital healthcare is on the rise across Asia, especially Southeast and East Asia. In Singapore, the National Electronic Health Record (NEHR) system was rolled out in 2011, and the country has begun transitioning to private clouds to store health data. Other countries such as South Korea and Thailand also have robust models for digital health). The concerns are compounded by the absence of a strong and comprehensive legal framework for data protection increasing the risk of privacy erosion and rights violations, the disproportionate impact of which is likely to be borne by marginalized groups and persons.

Here, it is pertinent to note recent events that allude to these possible threats that digitization may pose. While on the one hand, the mobile screening toolkit model adopted in Devarajeevanahalli is illustrative of the significant beneficial potential that digitization of healthcare possesses, a recent event that transpired at the All India Institute of Medical Science (AIIMS) reveals the heightened risk of data breach and the resultant rights violations owing to digitization in the absence of a comprehensive data protection framework. On 23 November 2022, several departments of AIIMS faced issues logging on to the e-hospital server and accessing patient records owing to a cyberattack where servers hosting the e-hospital database and laboratory data were hacked and corrupted [3]. The pandemic has increased dependence on digital systems. Cyberattacks have increased on hospitals and other medical institutions as hackers have realized the significant reliance of hospitals on digital healthcare systems to manage medical functioning and store and handle patient data [4]. Thus, privacy is a significant concern, and the possibility of patient data being compromised incentivizes pushback against implementing these digital systems on a larger scale, creating a greater reliance on them.

The current legal and regulatory landscape that governs digital health in India is scattered and ambiguous [5]. Moreover, there is limited legal scholarship on digital health in India. This is particularly challenging given the vast scope of digital health, covering various aspects of service delivery, data aggregation and processing, business models and technological advancements, leading to the fragmentation of the regulatory system. Unsurprisingly, there is skepticism around the large-scale digitization of healthcare in India due to the potential for mismanagement and misuse of data and leaks and data usage by private sector stakeholders. Privacy supporters have disagreed with policy steps by the government. Prasanth Sugathan, legal director of SLFC.in, a digital civil rights group stated, “the absence of a data protection law should not be an excuse to conduct such exercises affecting the rights of citizens. The fact that citizens agreed to provide their data for controlling the pandemic should not result in this data being used for other purposes without express and informed consent from the citizens [6].” Additionally, some government officials have criticized the Digital Personal Data Protection (DPDP) Bill, the first draft of which was circulated by the government on 18 November 2022, for giving the government power to exempt any of its agencies from compliance. Former Supreme Court justice B. N. Srikrishna called the Bill “a puppet of the government” and stated, “there is power to exempt all government institutions from any or all provisions of the law. That is a clear invitation to the executive to act arbitrarily [7].” In a country where information gathering, data collection, processing, and use are highly fragmented between the public and private sector and where out-of-pocket expenditure for healthcare is high amongst the population, the government must develop a clear data-protection framework as well as ensure that sufficient infrastructure, personnel, and facilities are available to maximize access to affordable, user-friendly, and streamlined digital healthcare. 

In this paper, I critically explore the legal and ethical challenges in regulating digital health. I highlight these challenges in the move towards digitization of healthcare in India by referring to the existing literature, laws, and policies introduced by the state. The first section of the paper gives an overview of the definition of Digital Health and is followed by a comprehensive analysis of the existing legal and policy framework in the second section. The paper then delves deeper into the two key concerns of informed consent and privacy that emerge from an analysis of the existing legal and policy framework before moving on to the concluding section, which discusses the recommendations for infrastructural and legislative changes that are prerequisites for realizing the potential of digital health in India [8].

## 2. Research Materials and Methods

A comprehensive review of the literature on the legal and policy developments in India was conducted to map the existing laws, guidelines, and regulations concerning privacy protection, data regulation, and digital healthcare provision mechanisms as introduced by the Government of India. The desk review of literature also included any reports issued by research institutions, journal articles, scholarly engagement with the issues and news reports and other relevant material to substantiate the review and analysis. 

## 3. Defining Digital Health

‘Digital Health’ refers to the growing convergences of digital technologies with healthcare delivery. The WHO has defined digital health as “a broad umbrella term encompassing eHealth, as well as emerging areas, such as the use of advanced computing sciences in ‘big data’, genomics and artificial intelligence” [9]. Digital Health, therefore, includes the “tools and services that use information and communication technologies (ICT) for purposes connected to health”, which may include improving outcomes of treatments for patients, diagnosing accuracy and closer monitoring of chronic diseases [6].

The digitization of healthcare involves two key components: the use of technology to deliver healthcare services and the digitization of medical data. The use of technology could include telemedicine, enabling patients to receive medical care without physical access to a healthcare professional or facility [2]. It could also include robot-assisted surgery and has, amongst other changes, led to an amplified focus on the use of artificial intelligence (AI) in various aspects of healthcare service delivery, including predicting, diagnosing, and treating diseases and conditions [10]. The digitization of medical data includes the formation of Electronic Health Records (EHR), a digital version of a patient’s health records that would allow doctors to view patients’ complete medical history, regardless of where and when the data were collected and can significantly streamline medical services [11].

The use of AI mechanisms that have data mining and pattern recognition capabilities can be successfully used to look at “symbolic models of diseases”, analyzing their relationships with patient symptoms to help with diagnosis, treatment, medical protocol development, drug development, and patient monitoring [12]. AI mechanisms can perform tasks that are traditionally carried out by humans in a quicker, more cost-effective manner, with the potential to “reinvent—and reinvigorate” modern healthcare [13] (Some of the tools that leverage AI in the field of medicine and healthcare include the use of virtual assistants, which have the potential to help people with Alzheimer’s Disease with their daily activities, technologies like MelaFind in the USA that uses infrared light to analyse pigmented lesions, assisting with preliminary skin cancer diagnoses, robotic assisted therapy that is used for rehabilitation of patients during stroke recovery and Caption Guidance, an AI software, which can help medical professionals capture echocardiographic images of patients’ hearts that can form the basis for diagnoses. *See* Castelo, M. (2020). The Future of Artificial Intelligence in Healthcare, *HealthTech*. https://healthtechmagazine.net/article/2020/02/future-artificial-intelligence-healthcare. (Accessed on 20 January 2023). 

One of the key transformative changes that a digital model of healthcare has the potential to implement is that of making healthcare access more rights based. To elucidate, as per the PANEL principles developed in public health literature, the principles of participation, accountability, non-discrimination, empowerment and legality are the pillars of ensuring access to healthcare services within a rights-based framework [14]. A rights-based approach converts human rights principles and standards into practice by translating rights from purely legal instruments to effective practices and policies on the ground. A rights-based approach to healthcare is based on the right to life, security, freedom from inhumane and degrading treatment, equality and non-discrimination, autonomy, privacy, and confidentiality, as enshrined in international and regional instruments, as well as national constitutions, laws, and policies. It foregrounds the principle of autonomy as essential to ensuring that access to healthcare is within a rights-based framework. The right to autonomy in making health decisions derives from the fundamental human right to liberty and is intrinsically connected with many fundamental human rights, such as liberty, dignity, privacy, security of the person, and bodily integrity. These rights form the basis for asserting individual decision-making in relation to health services and health care with respect to informed consent and confidentiality [15]. However, a rights-based approach also suffers from limitations as the concept of rights is essentially contextualized within a universal, ‘Western’ framework, and the application of this framework without due consideration to the socio-economic, cultural, and political specificities is likely to reproduce existing hierarchies than dismantle them [16]. Therefore, a conscious effort must be made to develop rights-based models of healthcare delivery that are indigenous and based on specific and clearly articulated socially and culturally relevant objectives. 

Arguably, the digitization of healthcare has the potential to allow greater access to medicine and healthcare services for people currently excluded from the health system or those who face several barriers to accessing quality healthcare. 

## 4. Legal and Policy Framework 

The project of digitization must be backed by a comprehensive legal and regulatory framework to harness the immense potential that digital healthcare possesses without compromising the rights-based approach toward access to healthcare. As noted above, the legislative and regulatory framework in India, in its current form, suffers from significant gaps and lack of clarity owing to the multiple disaggregated laws and policies. Each related legislation has been listed in Table 1. and will be discussed in turn in this section.

### 4.1. Information Technology Act and Rules

The Information and Technology Act, 2000 (“IT Act”), The Information Technology (Reasonable Security Practices and Procedures and Sensitive Personal Data or Information) Rules, 2011 (“Data Protection Rules”), and the Information Technology (Intermediaries Guidelines) Rules, 2011 (“Intermediary Guidelines”) govern a key element of digital health, namely the constant exchange of information between the patient and the service provider. Rule 3 of the Data Protection Rules defines Sensitive Personal Data or Information (“SPDI”) of a person to mean such personal information which consists of information relating to passwords; financial information; physical, psychological, and mental health conditions; sexual orientation; medical records, history, and biometric information. Therefore, when a Body Corporate (Section 43A of the Information and Technology Act, 2000, defines “body corporate” as any company and includes a firm, sole proprietorship or other association of individuals engaged in commercial or professional activities.) collects, stores, transfers, or processes such information, certain requirements under the Data Protection Rules are triggered. 

Consent is one of the key requirements under the Data Protection Rules, with Rule 5(1) stipulating that body corporates or individuals acting on their behalf are required by law to obtain a patient’s consent in writing before they can use the patient’s data. Patients must be informed of the collection of their data, the purpose of its use, and whether it will be transferred to any third parties, along with the contact details of the agency collecting the information (Rule 5(7) of The Information Technology Reasonable Security Practices and Procedures and Sensitive Personal Data of Information Rules, 2011). Body Corporates are mandated to allow users to have the option to withdraw their consent or modify their information if need be (Rule 4(1) of The Information Technology Reasonable Security Practices and Procedures and Sensitive Personal Data of Information Rules, 2011). Body Corporates are also required to have privacy policies in place published on their websites (Rule 7 of The Information Technology Reasonable Security Practices and Procedures and Sensitive Personal Data of Information Rules, 2011) and require prior permission before disclosing any patient’s SPDI to a third party. In cases where the SPDI is being transferred, the body corporate transferring the SPDI must ensure the receiver of the SPDI has adequate security practices in place in addition to obtaining the consent of the provider of information for such transfer (Section 2(w) of the Information Technology Act, 2000 defines an intermediary as any person who on behalf of another person receives, stores, or transmits that record or provides any service with respect to that record and includes telecom service providers, network service providers, internet service providers, web hosting service providers, search engines, online payment sites, online auction sites, online market places, and cyber cafes).

The Data Protection Rules also mandate the implementation of reasonable security practices and procedures to keep the SPDI secure. This requirement is fulfilled if the Body Corporate conforms to the international standard IS/ISO/IEC 27001 on “Information Technology—Security Techniques—Information Security Management System—Requirements” or similar standards that are approved and notified by the Central Government. At the time of writing, no such standards have been notified [6].

For health aggregators and m-Health platforms, Indian law treats them as ‘intermediaries’ under the Intermediary Guidelines and subsequent amendments (Rule 3 of the Information Technology (Intermediary Guidelines) Rules, 2011). Intermediaries have basic standards for due diligence mandated by the Guidelines (Section 79 of the Information Technology Act, 2000) and are also exempt from liability in certain cases (One relevant case is that of Shreya Singhal v. Union of India, which challenged the constitutionality of Section 79 of the IT Act, limiting liability of intermediaries, as well as the Intermediary Guidelines as vague provisions granting discretion to the intermediary regarding unlawful/offending material. The Supreme Court read down the impugned statutory provision and Rules, stating that intermediaries must receive a court order or notification from a government agency requiring them to remove specific information. (Shreya Singhal v. Union of India, AIR 2015 SC 1523.)), provided that the role of the intermediary is limited to providing access to a communication system over which the information is hosted or stored, and the intermediary has complied with the due diligence requirements prescribed.

### 4.2. DNA Technology (Use and Application) Regulation Bill

The DNA Technology (Use and Application) Regulation Bill, 2019, was introduced in July 2019 as a proposed legislation for the “regulation of use and application of DNA technology for the purpose of establishing identity of missing persons, victims, offenders, under trials and unknown deceased persons [17].” The primary purpose of the proposed law is the expansion of applying DNA-based forensic technologies to support and strengthen the justice delivery system of the country by providing for mandatory accreditation and the regulation of DNA laboratories across the country [17]. The bill called for enabling the application of DNA evidence in the criminal justice system and setting up infrastructure, such as National and Regional Data Banks, to assist in forensic investigations [17].

The bill was referred to the Parliamentary Standing Committee on Science and Technology [18]. The committee released a report that highlighted concerns about how DNA profiles collected under the proposed law could reveal sensitive personal information, which in turn could result in caste or community-based profiling of individuals [18]. Further, the bill came under fire for potentially compromising the right to privacy of individuals, with concerns that there would be inadequate safeguards at Data Banks to store DNA profiles safely [18].

### 4.3. Drugs and Cosmetics Act and Rules

The Drugs and Cosmetics Act, 1940 (“D&C Act”), along with the Drugs and Cosmetics Rules, 1945 (“D&C Rules”), regulate the manufacturing, import, sale, and distribution of drugs in India (“All devices including an instrument, apparatus, appliance, implant, material or other article, whether used alone or in combination, including a software or an accessory, intended by its manufacturer to be used specially for human beings or animals which does not achieve the primary intended action in or on human body or animals by any pharmacological or immunological or metabolic means, but which may assist in its intended function by such means for one or more of the specific purposes of—(i) diagnosis, prevention, monitoring, treatment or alleviation of any disease or disorder; (ii) diagnosis, monitoring, treatment, alleviation or assistance for, any injury or disability; (iii) investigation, replacement or modification or support of the anatomy or of a physiological process; (iv) supporting or sustaining life; (v) disinfection of medical devices; and (vi) control of conception.” (See Ministry of Health and Family Welfare, Notification, 11 February 2020, https://cdsco.gov.in/opencms/opencms/system/modules/CDSCO.WEB/elements/download_file_division.jsp?num_id=NTU0OA==) (Accessed on 20 January 2023)). The Medical Devices Rules, 2017 (“MDR”), framed under the D&C Act, established comprehensive quality requirements to be followed by marketers, importers, manufacturers, and sellers of notified medical devices. The D&C Act and MDR ensure the quality and safety of notified medical devices at all levels of the supply chain by enforcing a mandatory license requirement to be fulfilled before undertaking any commerce in notified medical devices, pursuant to quality checks.

Until 11 February 2020, the government had regulated or notified 37 categories of medical devices as drugs. On 11 February 2020, the government exercised its powers to notify one or more categories of medical devices as a “drug” to inform a new definition of medical devices. As per the notification, effective 1 April 2020, the medical devices that fall under the definition as notified will be regulated as a “drug” under the D&C Act and the MDR [19]. Therefore, the D&C Act and Rules outline the drugs that can be sold only via prescriptions issued by registered medical practitioners and those that can be sold over the counter, greatly impacting e-Health and m-Health platforms. Further, this has implications for the functioning of e-pharmacies that provide medicines via online portals to patients and customers.

### 4.4. The Clinical Establishments (Registration and Regulation) Act

The Clinical Establishments (Registration and Regulation) Act aims to regulate all clinical establishments in India (Since March 1st, 2012 the Act has taken effect in Arunachal Pradesh, Himachal Pradesh, Mizoram, Sikkim, and all Union Territories except the NCT of Delhi. The States of Uttar Pradesh, Uttarakhand, Rajasthan, Bihar, Jharkhand, Assam, and Haryana have also adopted the Act under clause (1) of article 252 of the Constitution. See “The Clinical Establishments (Registration and Regulation) Act, 2010.” (2015). Ministry of Health and Family Welfare, Government Of India, http://www.clinicalestablishments.gov.in/cms/Home.aspx (accessed on 20 January 2023)). The act is not automatically applicable to all Indian states, but state governments may adopt the regulation on their own volition [20]. The act aims to ensure uniformity and quality in the healthcare service received across the country.

The act is applicable to all types of clinical establishments, except for those owned or managed by the Armed Forces. The central government established a National Council of Clinical Establishments to determine the minimum standards of health care by a clinical establishment, classify them into categories, and maintain a national register of clinical establishments [21]. Section 38 of the act requires every state government to supply the central government with a digital record of all clinical establishments in the state, requiring each clinic to have a digital presence and digitize all patient records in order to receive registration [21]. Standardizing the digitization of patient records provides evidence that partially proves to the government that the clinic complies with the minimum regulations required for registration.

### 4.5. e-Health India

The Ministry of Health and Family Welfare (MoHFW) started various ‘e-Gov’ initiatives in the healthcare sector, calling itself ‘e-Health India,’ focusing on the use of ICT in healthcare. Under e-Health India, the MoHFW proposed the National eHealth Authority (“NeHA”) in 2015 as a “promotional regulatory and standards-setting organization in Health Sector [22].” The goal of NeHA was to “ensure development and promotion of eHealth ecosystem in India for enabling the organization, management and provision of effective people-centered health services to all in an efficient, cost-effective and transparent manner [22].”

In 2017, the MoHFW released the National Health Policy (“NHP”) with the goal of universal access to healthcare, advocating for the application of digital health initiatives in that regard [23]. The NHP has recommended the establishment of a Federated National Health Information Architecture to link public and private health providers consistent with Metadata and Data Standards (MDDS) and EHR and a National Health Information Network by 2025 [23]. 

Under its principal initiative of ‘Digital India’ [24], the Government of India launched the National Digital Health Mission (NDHM) in 2020, a nationwide campaign to create an open digital ecosystem that aims to improve the efficiency, effectiveness, and transparency of health service delivery. Under the mission, every ‘citizen’ will be given a unique Health ID which will be created by using basic details of the individual (such as a mobile number) or an Aadhaar number, and a unique biometric-based identification number will be issued to an individual [25]. Additionally, there will be a comprehensive repository called the Health Facility Registry that will contain information about all the healthcare facilities (both private and public) in India. (The features of the blueprint include “...a 5-layered system of architectural building blocks, Unique Health Id (UHID), privacy and consent management, national portability, EHR, applicable standards and regulations, health analytics and above all, multiple access channels like call centre, Digital Health India portal and MyHealth App.” See Ministry of Health and Family Welfare, National Digital Health Blueprint (2019).). The NDHM has been launched to address the lack of patient records and data in the Indian healthcare system. It is led by the National Health Authority (NHA) and is supposed to be a major step towards achieving a citizen-centric healthcare system and unique health IDs that will store the digital health records of individuals. Presently, hospitals in India fail to keep records of basic parameters, such as the date and time of consultation and diagnosis. 

The NITI Aayog proposed the National Digital Health Blueprint (“NDHB”) in January 2020, establishing the framework for a future health system that aimed to create a framework for the National Digital Health Ecosystem and be implemented through the NDHM. Centralized digital health records were first sought to be adopted in 2018 for the Ayushman Bharat-Pradhan Mantri Jan Arogya Yojana (AB-PMJAY) under the NHA. However, the government of India expanded the scope of digitization of health records to a national level. Consequently, the NDHB was released for public consultation in October 2019 [26]. Therefore, the NDHB is a step towards standardizing the use of e-Health in India and has the potential to create a framework for implementing widespread digitization of healthcare throughout the country. 

### 4.6. Health Data Management Laws and Regulation

The Government of India introduced the Health Data Management Policy in 2020, claiming it would protect citizens’ health data by regulating their collection and storage [27]. The objectives of the draft policy included creating a system of digital personal and medical records, which would be voluntary and based on the consent of individuals and allowed for the secure exchange of health data with enough privacy safeguards [27]. The 2020 draft of the policy received criticism for allowing private entities to have access to health data, and the NHA released a revised version of the Health Data Management Policy in April 2022 [28]. Under the same, the changes related to the creation and issuance of the Ayushman Bharat Digital Account (ABHA), non-consensual processing of data, and data localization are among the key things that have been addressed [29].

On the national level, the government has set up an Integrated Health Information Platform through the MoHFW to enable the establishment of EHRs of citizens on a pan-India basis and the integration and interoperability of the EHR over a centralized, accessible platform [30]. EHR Standards were released in 2016 to create a standard model that could be applicable across states and amenable to interstate coordination [30]. A Personal Health Record Locker by the name of “MyHealthRecord” has also been introduced by the government [30].

Acting on the vision originally propounded by NeHA, the MoHFW introduced a draft bill titled the Digital Information Security in Healthcare Act (“DISHA”) in 2018 for public comment. It intended to grant control over digital health data to owners and to provide the framework for the Ministry to utilize patient data in programmes in a secure manner [31]. DISHA sought to establish NeHA and facilitate the online exchange of patient information [32]. In 2019, the MoHFW submitted DISHA to be subsumed under the ‘Data Protection Framework on Digital Information Privacy, Security & Confidentiality’ Act, which was drafted by the Ministry of Electronics and Information Technology (MeitY) [33]. This draft was eventually introduced as the Personal Data Protection Bill, 2019 (PDP Bill), in the Parliament. Many lawyers and activists penned critiques of the PDP Bill, particularly raising concerns related to violations of the right to privacy. The PDP Bill required sensitive personal data, including medical records, to be processed with the explicit consent of the data owner, also putting in place data localization requirements, by which all data fiduciaries must store at least one copy of this personal data in a data center located in India [6]. This alarmed privacy advocates and tech giants, including Meta, Google, and Amazon. Under the PDP Bill, the management of sensitive personal data would be restricted, while the government would have broad powers to access it (Submissions made by organizations such as the Observer Research Foundation (ORF) pointed to the need for clearer standards regulating government access to anonymized data. Furthermore, consultations by the ORF also brought up concerns over the “the lack of additional safeguards, rules or regulations related to secondary use, i.e., use for purposes other than what was envisaged at the time of collecting the data.” (See ORF Technology and Media Initiative, (2020) The Personal Data Protection Bill 2019: Recommendations to the Joint Parliamentary Committee, ORF Special Report No. 102. Available at: https://www.orfonline.org/research/the-personal-data-protection-bill-2019-61915/ (Accessed on 20 January 2023)); As Anirudh Burman stated, data is often collected using long forms or contracts that users either do not understand or do not take the time to read through. In fact, Burman noted that the Justice Srikrishna Committee—which drafted the first version of the Bill—itself emphasized that this concept of consent was ‘broken’. See Burman, A. The draft Personal Data Protection Bill is flawed. HINDUSTAN TIMES, (2019), https://www.hindustantimes.com/analysis/the-draft-personal-data-protection-bill-is-flawed-opinion/story-NoUUk81zW7d8Xniarn9tML.html (Accessed on 20 January 2023). One of the primary challenges related to the bill was the reliance on user consent [34]. The bill also mandated that data designated as critical data by the government could only be stored in India [34]. This included financial, health, and biometric information [6]. Unique digital IDs would provide a single source of health information for the patients, government, and healthcare providers through interoperable health records. Research shows that the adoption of EHRs leads to an increase in the quality of care received by patients while decreasing the costs [35] of these services. 

The PDP Bill was withdrawn by the Government in August 2022 following several critiques. The Government of India subsequently released a draft DPDP Bill in November 2022 [36]. Supposedly, the DPDP Bill provides a more comprehensive framework than the PDP Bill. Its key principles cover rightful usage, resolute dissemination, and relevant data collection, amongst other principles of the data economy. Its provisions include regulations on the processing of data, the prevention of data breaches, the protection of children’s data, and data grievance redressal. Most notably, Clauses 6, 7, and 8 cover consent, a concept that is emphasized in this bill [37]. However, the bill’s conditions surrounding consent are concerning. The bill has received mixed responses from legal experts, with some expressing skepticism for giving the government excessive authority in matters relating to consent and privacy protection, which in effect diluted the attempts made to enact a data protection framework [38]. For example, Clause 8, which covers the provision of “Deemed Consent”, states that the data principal is presumed to have given consent when it is for “any fair and reasonable purpose “ as may be prescribed [37]. This is extremely broad and allows the government to deem consent whenever they determine that it is “fair”. Further, Clause 8 takes away consent from the data principal when “the state or its agencies need to perform any function under any law” or “provide any service or benefit to the Data Principal [37].” The phrasing of these statement makes the data principal a passive subject. Perhaps the most important thing to note about the DPDP Bill, is the fact that the government forms one of the largest data fiduciaries, processing the personal data of millions of Indians. This makes it important to place data management and protection rules at arm’s length from the government to ensure impartiality. Vesting the power of data protection to the government, which this regulatory framework does, becomes problematic (Section 43A of the Information Technology Act, 2000). The fact that the government itself would be subject to these rules creates a conflict of interest. 

The DPDP Bill also has dubious regulations regarding data protection. The bill does away with the category of sensitive personal data altogether and envisages significant amendments to the IT Act, particularly the omission of Section 43A of the IT Act, which states that organizations are liable to pay damages when they deal with any sensitive personal data and fail to maintain “reasonable security practices and procedures and thereby causes wrongful loss or wrongful gain [39].” Thus, there are no specific requirements to protect sensitive data sets, including health, financial information, biometrics, genetics data, and other particularly important matters. This form of personal data is typically afforded a higher degree of protection, requiring explicit consent and mandatory data protection impact assessments [40]. The removal of the distinction between sensitive personal data and other data creates risks when digitizing healthcare because health data are no longer considered sensitive data. The bill is currently at the stage of public consultations, and what comes into effect remains to be seen. Thus, at the present stage, the lack of an effective legal framework for consent for data collection in the country poses increased risks related to digitization. 

### 4.7. Telemedicine Practice Guidelines

Although telemedicine consultations (‘teleconsultations’) have been offered for many years in India, there was no statutory basis for the provision of these services, leading to bodies such as the Karnataka State Medical Council warning doctors in Karnataka against the provision of online consultation, with consequences including the cancellation of registration if found to be in violation [41]. On 25 March 2020, the day India went into a nationwide lockdown to curb the spread of the COVID-19 pandemic [42], the government published the Telemedicine Practice Guidelines, allowing teleconsultation by registered medical practitioners for the first consultation between doctors and patients remotely [43]. The guidelines were enacted “to give practical advice to doctors so that all services and models of care used by doctors and health workers are encouraged to consider the use of telemedicine as part of normal practice,” to be “used in conjunction with the other national clinical standards, protocols, policies and procedures [41].”

The guidelines require doctors to maintain the same standard of care towards a patient during a teleconsultation as they would during an in-person consultation and also cast a duty on patients to provide accurate information to medical professionals [43]. While doctors may use any medium (including telephone, mobile, WhatsApp, Facebook Messenger, mobile apps, or internet-based platforms) for consultations with patients, they are required to exercise their professional judgment to decide whether a teleconsultation is appropriate and in the interest of the patient [43]. The guidelines further clarify that in cases where patients are below the age of 16 years or incapacitated (due to mental conditions such as dementia or physical disabilities due to an accident), the teleconsultation can take place with the caregiver without the presence of the patient [43].

In accordance with the guidelines, all doctors providing teleconsultation will have to affirmatively identify themselves to the patient before starting every teleconsultation and display their registration number in all communications with the patient, as well as when they issue prescriptions [43]. If a fresh prescription for chronic diseases (such as asthma, diabetes, or hypertension) is to be issued, then the teleconsultation should be undertaken strictly via video, and prescriptions can be sent through electronic media such as email, WhatsApp, etc., as a photograph, scan, or digital copy of a signed prescription or an e-prescription [43]. There is no fixed format for issuing a prescription, and the guidelines only provide a recommended format [43].

These guidelines provide a clear set of dos and do-nots for doctors and cast obligations on doctors to maintain patient records for teleconsultation, including case history, investigation reports, copies of prescriptions, and proof of teleconsultations (The recent experience with the COWIN portal for COVID-19 vaccinations serves as an illustrative example. The government of India’s decision to mandate all individuals between the ages of 18–44 years to register in order to be administrated the vaccination was scrutinized by the Supreme Court of India, noting that lack of access to internet facilities given the digital divide in the country would create significant hurdles, especially for many in poor villages to obtain access to vaccines. In an interim order, the Court flagged the digital divide as a key concern that could have severe implications on the right to equality and the right to health if the vaccination policy was to rely exclusively on a digital portal to provide access to every individual in the country. (See order dated April 30th, 2021 in In Re: Distribution of Essential Supplies and Services During Pandemic, Suo Motu Writ Petition (Civil) Bo. 3/2021)). The information gathered would fall within the scope of the Data Protection Rules, prohibiting disclosure and transfer without the written consent of the patient. This would be in addition to ethical obligations to protect patient privacy under the Code of Conduct for medical professionals. The guidelines also grant the option for patients to address grievances stemming from violations therein before appropriate State Medical Councils. They also require patients to provide thorough and accurate information to their medical professionals.

The guidelines encounter complications surrounding consent, a lack of access to devices for consultations in marginalized communities [44], as well as the ethics surrounding obtaining consent from marginalized persons. When the teleconsultation is with a child below the age of 16 or with a person with mental or physical disabilities, the consultation may take place only with the caregiver. This is problematic, as the doctor cannot see the patient in person, and their judgement would be entirely based on the caregiver’s opinions. People with physical disabilities may not be mentally incapacitated and capable of interacting with their doctors themselves. Further, caregivers of children may not be able to provide accurate information, as there are things about their child’s health that the child will not tell them. Thus, though there is a more comprehensive framework surrounding teleconsultations following the pandemic, there are considerable challenges in the current legal framework.

An overview of the legislative and regulatory measures implemented by the government in support of the digitization of healthcare reveals there is a significant political will on the part of the government as far as the project of digitization is concerned. However, when one considers the sheer volume of data that such a leap toward digitization will generate, the current legal framework is lacking. While the issue of the abuse and misuse of data is a concern, there is a larger problem that precedes that, one that begins at the very stage of the collection of data. The principle of informed consent as a principle of medical ethics becomes extremely critical when looking at the digitization of healthcare, and this problem is further complicated by technological, linguistic, and other barriers and a lack of digital literacy. The expansive collection of laws and policies that have been discussed above have been designed without due consideration to the socio-cultural and infrastructural challenges that hinder access to healthcare. They are thus lacking in a rights-based approach towards the digitization of healthcare, an essential component for harnessing its immense potential to actualize universal access to healthcare. If there are consent architectures and data-sharing models that can be implemented in a manner that provides data subjects with control over their data and secures their privacy, and the potential for improving the quality of healthcare increases. This can only be undertaken if digitization is a slow and thought-out process that engages all stakeholders, not just patient or healthcare providers, but also persons working on privacy-related concerns, those working with Accredited Social Health Activist (ASHA workers), among others, to ensure there is universal information and access.

## 5. Identification of the Key Legal and Policy Challenges: Informed Consent and Privacy

The are several challenges relating to implementing digitization programs, with India’s healthcare infrastructure being short by almost half of what the WHO recommends as the optimal number of doctors, nurses, medical technicians, and healthcare facilities required to serve the population [45]. Under excessive strain for decades and without sufficient budgetary support, the country’s health system is overburdened by the tackling of infectious diseases (tuberculosis and neglected tropical diseases), as well as so-called lifestyle disorders (diabetes, stroke, and heart and neurological problems) [46]. Therefore, I have identified two key legal and ethical challenges, i.e., informed consent and privacy, as a focus of this paper. These will be discussed in turn.

### 5.1. Informed Consent

Consent is the expression of the right to autonomy, granting an individual the freedom to make decisions and to make active, conscious, affirmative, and voluntary agreements to engage in an activity or process [45]. Respect for autonomy is considered one of the key grounding principles underlying consent. It is, however, pertinent to note that although consent may be the best-recognized way to permit disclosures of private information, consent is often not informed or given under economic duress and thus does not provide sufficient protection to patients [47]. Therefore, it is imperative that the consent of the patient be informed to protect their interests. Further, it is important for consent to be maintained for all medical interactions, whether in-person or via telemedicine [28]. An individual’s right to autonomy and ability to exercise consent is mediated by a range of structural, political, legal, socio-cultural, economic, attitudinal, and capacity-related barriers. The United Nation’s International; Disability Caucus defines the term “informed consent” as “Informed decisions [that] can be made only with knowledge of the purpose and nature, the consequences, and the risks of the treatment and rehabilitation supplied in plain language and other accessible formats [48,49].” The definition ensures equal decision-making powers for persons with disabilities [49] and other marginalized individuals. Informed consent for treatment is based on the premise that individuals not only have a right to decide what happens to their bodies but that both medical goals and individual goals must be considered for treatment choices [50].

The concept of informed consent has matured over the years. The term surfaced in the 1950s and, over the ensuing 20 years, revolved around the obligation to disclose information to the patient. [51] From the 1970s to the present, the obligation of disclosure evolved into the necessity of the subject understanding and consenting to the treatment course on which they were about to embark [51]. The key elements of informed consent are disclosure of information, patient understanding, voluntariness, and authorization [51]. Informed consent can only occur through autonomy [50]. Informed consent implies the provision of comprehensive information to the concerned person whose data are being procured in this case, and in the context of healthcare in particular, informed consent is defined as the right and responsibility of every individual to make informed decisions about their health and care together with their healthcare providers. In order to facilitate this, healthcare providers must provide patients with the information and support that is evidence-based, culturally appropriate, and personalized [51].

The principles of self-determination and individual patient autonomy are at the foundation of informed consent. Autonomy, when defined within the human rights framework, is understood as the right of an individual to make free and autonomous decisions about their bodies, sexual and reproductive capacities, functions, and choices, free from any coercion or violence [15]. The right of autonomy is at the core of the fundamental rights to equality and privacy and is among the core human rights principles that must be accounted for in ensuring the realization of the health rights of all persons without any discrimination. Consent is the expression of the right to autonomy, granting an individual the freedom to make decisions and to make active, conscious, affirmative, and voluntary agreements to engage in an activity or process. Individual autonomy encompasses an individual’s right to control their choices, as well as the right of freedom from unwanted interference in that regard, only achieved if patients are sufficiently informed of their rights to refuse medical treatments (105 N.E. 92, 93 (N.Y. 1914)). Individual autonomy was advocated in Schloendorff v. Society of New York Hospital [52], wherein Justice Cardozo wrote, “Every human being of adult years and sound mind has a right to determine what shall be done with his own body.” In the Indian context, the importance of informed consent can perhaps be most clearly seen in the administration of the human papilloma virus (HPV) vaccine to *Adivasi* (Indigenous) girls in Andhra Pradesh and Gujarat in 2007. After the death of four indigenous girls, the administration of the vaccine immediately stopped. The girls and their caretakers were not given information to understand the purpose of the vaccine or its side effects [53]. Kalpana Mehta filed a public interest litigation claiming the vaccines were approved after 2007 when they were administered [54]. The lawyer Colin Gonsalves argued that these were human trials that violated adolescent girls [54]. Even if the vaccines were administered with the good intention of ending the prevalence of HPV amongst indigenous women, doing so without their informed consent is dehumanizing. Consent and communication are rights-based principles and must be emphasized when providing healthcare to marginalized persons. 

Melinda McCormick identifies the main concern with ‘informed consent’ as being that “subjects, to the degree that they are capable, be given the opportunity to choose what shall or shall not happen to them [55]”. For instance, when conducting research, one may provide information to survey respondents about the research, but it is difficult to confirm if they have adequately comprehended the implications of the research [55]. Indian ethicist Amar Jesani notes that at an institution referred to him for ethical consultation, patients who attempted to receive care at the hospital were asked to give written general consent stating they consented to the hospital using their medical records, including case papers, X-rays, ultrasound scans, and more for use in any research. The only compromise provided by the institution was that the patients’ names and likenesses would not be disclosed [56]. This blanket consent is justified by the fact that it is a cost-effective way for students to conduct research. It is also argued that the research contributes to the good of the public, improving patient care [56]. However, Jesani highlights that speaking for the public is problematic, as one cannot provide informed consent when they do not have specific information on what the consent entails [56]. This is particularly salient when patients are in a vulnerable position and need medical care.

Informed consent is one of the major requirements of the Data Protection Rules. The DPDP Bill also requires that data are only processed for lawful purposes [37] but adds that the contact details for a Data Protection Officer must be provided as a resource, the data principal has the right to request that their data be redacted or erased, and that the data fiduciary cannot monitor or target advertisements to children [37]. Further, under the NDHB, data are to be collected as per a consent framework that has been established for the collection and processing of health data [57]. The blueprint states that the data source is to be considered as the owner of the data, which shall exercise control over the nature of the data collected, with whom and how the data are shared, and the purpose and processing of data. Anonymized data may be accessed by the government to study disease prevalence and for framing policies. However, the blueprint is silent on what the consent framework would be like at the ground level and has been said to fall short of ensuring that patients provide informed consent with respect to the way their health data would be used [58].

The lack of a legal framework could impact access to data protection broadly. The rapid growth of e-commerce in India is proof that a comprehensive framework is needed to regulate online industries. Online payments have rapidly increased in the country, which necessitates that the government provides protections against criminals who perform scams, pushing for robust e-governance and a digitally empowered society [59]. This includes digital health. Digital health could also be impacted by a lack of government protections. For example, frontline health workers in Karnataka were supported by the assets of digital health, such as engaging, video-based health education material [60]. However, they were hindered by the lack of infrastructure, as digital initiatives have overlooked broader socio-cultural practices that influence the way frontline workers operate, and the digital framework has not adapted to their needs. For instance, community healthcare workers must combat persisting issues, such as language barriers, social stigmas associated with HIV status, and inter-personal power dynamics within communities, which continue to battle embedded hierarchies of caste and gender, among other issues [61].

A common problem with obtaining user consent for processing their data is the difficulty in explaining to the user in simple terms what data are being collected, how it is being processed, and whom it may be shared with. Therefore, in addition to obtaining user consent for collecting and processing data, a robust regulatory framework governing health records should be put in place. India’s ASHA workers, who are often residents of the poor and rural villages they serve, should be more highly valued parts of the healthcare system and provided training to be telemedicine coordinators in low technology and low resource areas [55]. For example, users may consent to websites using their “cookies” without realizing what this means and that it may track and collect data related to their online behavior, including a record of their website visits and activity. This information may then be provided to advertisers without their knowledge [62]. This problem is compounded in India due to low levels of digital literacy. The concerted push for large-scale digitization is confronted with the challenges that result from a lack of digital literacy among almost 90% of India’s population [63]. There is a further disparity when it comes to the level of digital literacy in urban and rural India, with the digital literacy rate being 61% in urban India as contrasted with 25% digital literacy in rural areas [64].

When dealing with marginalized persons particularly, the contours of informed consent become even more challenging to navigate. For example, in research focusing on adolescent girls with anorexia, it has been noted that the assumption that parents know what is in the best interest of their children does not always hold true [14]. Similarly, in the case of abortion laws in India, the requirement of parental or guardian consent has proven to be a major barrier to access to abortion and other reproductive healthcare services [65]. Further, given the lack of digital literacy in India, the process of obtaining informed consent may further entrench the asymmetries of power and social hierarchies that define the relationship between a medical practitioner and a patient. For instance, there is a widespread stigma against transgender and gender-variant persons in India receiving gender-affirming surgery or hormone therapy. Transgender persons must provide a certification from a psychiatrist affirming their dysphoria in addition to informed consent [65], indicating skepticism around a transgender person’s ability to determine whether they want to pursue gender-affirming treatments, thus taking away from their right of autonomy. It also provides the additional barrier of a psychiatrist’s referral, which many persons cannot afford [66]. One of the alternatives proposed is that of ‘ongoing consensual decision making’ where “the respondent is kept informed as to their vulnerability to potential dangers and the decisions regarding the research are made as a team” [67]. Digital health may be more widely accessible, but the model for informed consent is currently underdeveloped. Thus, it is essential to establish a comprehensive legal framework with set rules around obtaining informed consent within a rights-based framework, which would entail taking note of the structural inequalities that can determine the extent to which an individual can have autonomous decision-making power over their personal data [68].

This means that organizations must have standard measures ensuring that patients understand what data are being collected and why. With consideration for the unique needs of marginalized communities, including sensitivity towards their socioeconomic conditions and privacy rights, a rights-based framework can be implemented, which not only ensures that informed consent can be actualized in a manner that secures the rights of autonomy and protects the privacy and confidentiality of the person concerned. The challenges concerning privacy are discussed in the next section.

### 5.2. Privacy

Privacy has long been an integral part of medical ethics and dates to the Hippocratic Oath. The oath is based entirely on the physician’s duty to maintain privacy; however, it is pertinent to note that the oath rests entirely on patient privacy and makes no reference to the autonomy of the patient or the shared decision-making model of physician–patient collaboration [68]. As enumerated in the discussion on informed consent, the relationship between the physician and the patient was highly paternalistic, and patients were merely passive recipients of healthcare.

The modern definition of privacy incorporates individuals or groups determining for themselves when, how, and to what extent private information about themselves is communicated to others [68]. Although this understanding of privacy is more progressive than that of the oath, it still fails to incorporate all types of privacy. It can be said that privacy in e-medicine is of two types: informational and physical [69]. It is physical when digital health allows you to reduce the number of home visits by doctors and in-person clinic visits. However, with this comes the threat of informational privacy as patient medical data are being stored on the internet, which, to say the least, can be extremely unreliable and prone to tampering. What should be noted, however, is that not only are there at least two kinds of privacy, but these two kinds of privacy can come into conflict with each other, potentially requiring tradeoffs between them [69]. The Stanford Encyclopedia of Philosophy defines the most important kinds of privacy for maintaining data protection in digital healthcare to be “informational privacy” and “decisional privacy [69].” As stated above, informational privacy is concerned with maintaining data protection and security. The rise of health monitoring devices, electronic copies of medical records, and DNA testing, among other things, mean that individuals lack substantial control over their medical privacy [69]. Meanwhile, decisional privacy in medical contexts is commonly used to denote autonomy in health-related decision making. For example, birth control and abortion are both medical practices that require reproductive autonomy and decisional autonomy. Additionally, the right to refuse medical services is an aspect of decisional privacy [69], emphasizing the importance of informed consent.

There are certain ethical justifications for confidentiality and privacy, including a consequentialist justification and a deontological justification [69]. The consequentialist justification states that privacy and confidentiality have instrumental value, as they promote important social goals, including the enhancement of individuality, self-determination, and the freedom to cultivate intimate relationships free from public life [69]. In the health realm, this justification has ethical significance as it encapsulates the promotion of the patient–provider relationship and the protection of the patients’ social status. It is also instrumental since it helps maximize good patient care and minimize potential patient harm. The deontological justification, on the other hand, justifies privacy and confidentiality in terms of respect for persons, which is grounded in the fundamental principle of autonomy [69]. Privacy and confidentiality help protect the moral agency of patients by allowing them to live their lives as they choose, resting on the intrinsic value and dignity of autonomous persons rather than their instrumental value and the ends served by them [69].

Privacy and confidentiality, though similar, have certain inherent differences [69]. Confidentiality is relational and must include two persons: one who discloses private information to another with the expectation that the same will remain confidential—e.g., with respect to doctors and patients [69]. Compromising the privacy of a patient will result in a trust deficit for healthcare service providers and reluctance to share private information, particularly in such instances where the information can be stigmatizing, as may be the case with information pertaining to risky sexual activity, HIV status, chimerical or substance abuse, or mental health problems, among other things [69].

Privacy is integral to all forms of healthcare, including digital healthcare. With the reliance on technology in modern India, there needs to be even more protection for patient medical data. Safeguards should be put in place controlling who has access to these data, and patient information should only be released with the patient’s explicit consent.

#### 5.2.1. Privacy as a Fundamental Right in India

The concerns around the protection of personal data and the right to privacy in India are further complicated by the judicial discourse on the issue. Privacy has been intensely contested and litigated in India since the 1950s (In 2017, the Supreme Court ruled in a landmark nine-judge bench decision that the right to privacy is a fundamental right in India. The Puttaswamy decision was the outcome of a constitutional challenge to the validity of the Aadhar biometric identity scheme. Prior to this, the Supreme Court had held in MP Sharma v. Satish Chandra and Kharak Singh v. State of UP that privacy was not a fundamental right. Subsequently, in Gobind v. State of MP, the Court recognized a constitutional right to privacy but affirmed that it would still be subject to restrictions. Puttaswamy overruled both MP Sharma and Kharak Singh to hold that privacy is a fundamental right protected by the Constitution), and several cases have established that the right could be reasonably restricted based on state interests. However, the Puttaswamy v. Union of India judgment in 2017 [70] marked a paradigm shift in this discourse, with the Supreme Court holding that the right to privacy is a fundamental right in India. The nine-judge bench judgment put forth a three-tier test for checking whether legislation violates the right to privacy. The first relates to legality, which means it must be a valid law. The second relates to need, which means there must be a legitimate state aim in enacting that law. Third is the test of proportionality, where the onus is on the state to demonstrate that the nature and extent of the curtailment of a right are proportionate to the legitimate aim sought to be achieved. Additionally, the current Chief Justice of India, Justice Chandrachud (on behalf of himself and three other justices), stated that privacy is not surrendered simply because an individual is in the public sphere. The court found that privacy is intrinsic to a life with dignity [71]. Along with holding that privacy is a fundamental right, the judgment also declared informational privacy to be a subset of the right to privacy, with information about a person and a right to access that information needing protection (In considering the right to privacy, there would be other considerations, on which, the Court observed as follows: “Formulation of a regime for data protection is a complex exercise which needs to be undertaken by the State after a careful balancing of the requirements of privacy coupled with other values which the protection of data sub-serves together with the legitimate concerns of the State”) [71,72]. Therefore, two points emerge that are critical: firstly, the primary value that any data protection framework serves must be that of privacy; second, such a framework must not overlook other values, including collective values (See the decision in the case of Martin F. D’Souza v. Mohd. Ishfaq, decided by the Supreme Court in 2009, where it was ruled that prescriptions must only be provided to patients over the phone in the case of emergencies and not as an ordinary practice. See also the decision in Zaheer Ahmed v. Union of India and Ors. (2018) where a petition seeking a ban of sale of drugs online due to a patient having obtained certain Schedule X drugs through an e-pharmacy without any prescription led the Delhi High Court to impose an interim ban on sale of drugs till 8 January 2019).

Despite this judgment, however, privacy is not an absolute right in India. In 2018, the court ruled that the Aadhar Act—which had formed the basis of the 2017 decision—was constitutionally valid as the intrusion of privacy was proportional to the object of the legislation; the conferment of social entitlements [71]. Here, Justice Sikri, the author of the majority opinion of the bench, laid down a four-fold test to determine proportionality; first, that a measure restricting a right must have a legitimate goal; second, that such a measure must be a suitable means of furthering the goal in question; third, that there must not be a less restrictive but equally effective alternative measure that can be adopted; and fourth, that the measure shall not result in a disproportionate impact on the right holder. Testing the constitutional validity of the Aadhaar Act on this four-legged threshold of legitimacy, suitability, necessity, and balance, the majority found that the act in question, barring some of its provisions, satisfied the four-fold test. The court also found that given that there was no less restrictive but equally effective system, the Aadhar, a unique, biometric-based identity system, also met with conditions of necessity and was thus constitutionally valid [71].

The Indian judiciary has maintained an outlook of skepticism regarding the digitization of health, especially in the field of telemedicine [73]. The orders of the Supreme Court on the fundamental right to privacy of persons, as well as the judgments from various High Courts [73] pertaining to telemedicine and e-Health, show that the Indian judiciary is cognizant of the lack of sufficient regulations pertaining to the digitization of health in the country. In this background, courts have been restrictive in allowing diagnosis, treatment and, most importantly, the prescription and sale of medication without in-person consultation between RMPs and patients. 

#### 5.2.2. Health Data Protection and Privacy Concerns

A significant challenge in digitizing health is the maintenance of privacy, in view of the ambiguities and gaps in the legal frameworks surrounding the same, as the current framework may fall short in governing health data at a large-scale level. The IT Act and Data Protection Rules are not specific to healthcare data protection, providing only a basic level of data protection by mandating that the data owner is informed about the manner of use of the data. Further, the International Standard Organization’s (ISO) 22600:2014 Health Informatics—Privilege Management and Access Control [57]—is recommended by the NDHB. However, it is pertinent to note that implementing ISO standards may be difficult for most healthcare providers in India and monitoring data protection compliance at the healthcare provider level may also prove to be a challenge.

As discussed earlier, the proposed DPDP Bill is highly unlikely to protect privacy adequately. Thus, the DPDP Bill most strikingly places few restrictions on the government’s use of personal data [74]. On paper, data protection rules apply to government agencies. However, the central government is one of the largest data fiduciaries and would have wide-ranging powers to exempt any public entity from the requirements if it is “reasonable” under the guidelines of deemed consent. The DPDP Bill is therefore strengthening the state’s role in the data economy and consequently increasing the state’s power for surveillance without any checks and balances (Section 43A of the Information Technology Act, 2000).

The European Union’s General Data Protection Regulation (“GDPR”), effective in 2018, aimed to protect the informational privacy of individuals by creating a framework that regulates how businesses collect and use personal data. The current DPDP Bill in India departs from this international standard and does not grant the category of sensitive personal data more protection than other forms of data [37]. The GDPR is much more thorough in terms of providing the right to transparency and protection than the DPDP Bill [75]. As seen earlier, a variety of underlying harms to patients may result from unwanted disclosures of sensitive health data, which can result in economic harm, such as the loss of employment, insurance, or housing [75].

In India, where only half the population uses the internet [76], the nationwide digitization of health is a rather contentious policy. The health data management policy of the NDHM covers “personal and sensitive personal data [77]”. Data are to be stored at three levels: Central, state, or Union Territory and health facility [77], and the ownership of personal data lies with the individual [77]. However, anonymized data, in an aggregated form, may be made available for research, statistical analysis and policy formulation [77].

In addition to the legal and institutional context of how data collection would take place, the limitations of India’s digital health infrastructure significantly raise the likelihood of discrimination in the course of use, processing, and disclosure of data. It may be used, hypothetically, to concentrate unfairly on a certain part of the population and may produce a misleading perception of the existence of or transmission of a disease [78]. It is, therefore, important to understand how a policy mechanism that aims at boosting the efficacy of the healthcare sector of a nation could disproportionately affect a particular section of the society and be discriminatory towards particular communities.

Data manipulation is also a potential concern with NDHM (The Tablighi Jamaat is a transnational Ismalic Missionary Movement that aims to reach out to ordinary Muslims and revive their faith, particularly in matters of rituals and encouraging fellow members to practicing their religion as per the directives of Prophet Mohammed). As recently witnessed in the case of Tablighi Jamaat [79], unsystematic and unaccountable records of data can tilt the statistics in a way that benefits a particular section of the society and thereby drive policy decisions in an inequitable manner. In March, 2020 during, the COVID-19 pandemic, there was an international congregation of the Tablighi Jamaat from different countries that was taking place in Nizamuddin Markaz, New Delhi. This preceded the restrictions that were imposed by the government in the wake of the Clovis-19 pandemic and by the time the restrictions were put in place, several members of this congregation had travelled beyond Delhi to other parts of the country. With an increase in the number of positive cases came a stricter scrutiny by the government which labelled the Jamaat as a super-spreader event and the resultant violence and religious discrimination against Muslims all over the country was an immediate consequence of this act by the government [80]. Owing to the fact that NDHM does not talk about the mechanisms of data collection and modes of documentation of people’s health records, [81] the database may only show statistics and data, whether favorable or not, in a selective and limited manner. Although governmental ministries are stipulated as primary users of data, one cannot deny corporate involvement and vested interests associated with data brokering and mining of such personal health information (The Tablighi Jamaat is a transnational Ismalic Missionary Movement that aims to reach out to ordinary Muslims and revive their faith, particularly in matters of rituals and encouraging fellow members to practicing their religion as per the directives of Prophet Mohammed). In addition to the aforementioned privacy and data surveillance concerns, NDHM also fails to address socio-cultural, economic, geographical and political barriers to its proposed plan of digitizing healthcare. Further, the NDHB’s aim of a federal database is said to be problematic, as in the absence of addressing data flow, the usage of these data in the data structure and the digital divide across health workers and patients will lead to excluded data and “service poisoning”.

Telemedicine (and other digital health initiatives) in India face major challenges from issues such as unreliable electrical supply, inadequate internet bandwidth, video distortions, and software malfunctions [45]. Further, in India and across the globe, there is an observed reluctance by both patients and doctors to fully adopt eHealth [45]. Heavy documentation pertaining to patients, rather than ‘easing doctors’ burdens’, could disadvantage doctors in India, as an average Indian doctor sees 40–60 patients in a day and is thus much more treatment-oriented (as opposed to record-oriented doctors, say, the United States) [3]. The need for meticulous documentation has seen strong resistance from the medical fraternity in the past, as seen during the proposal of The Clinical Establishments (Registration and Regulation) Rules, 2012, which mandated that all hospitals and clinics maintain electronic health records of patients. The Indian Medical Association led a nationwide strike against this, as many small and medium hospitals do not have the capacity or time to store digital patient records [82].

In this context, the lack of robust data protection laws is a significant concern. Privacy in the realm of digital health comes with great complexities. This is because information about a person’s health can be extremely intimate, e.g., whether they are HIV positive or have had an abortion. Further, individuals whose personal data are collected can be amongst the most vulnerable in society, such as HIV-positive persons, transgender and gender-variant persons, adolescents, etc. An intrusion into the privacy of such people can be even more harmful and arguably calls for even greater protection [83]. There are a number of specific means by which personal health information (PHI) can be compromised: (i) cookies and spyware, which allow unauthorized individuals to monitor computer use and track online activities, (ii) hackers can gain illicit information, iii) information can be transmitted to unauthorized individuals or the World Wide Web accidentally, (iv) poorly designed security measures and the inadequate training of staff, and (v) the fact that entities that hold patient information have incompatible security measures, making it more likely that the information will fall into the wrong hands [83].

It is telling that in India, the only actors that seem to be unquestioningly enthusiastic about health digitization are businesses-insurance companies, and private digital health services. This reveals that in the absence of an adequate legal framework for data protection and regulation and increased distrust in the public healthcare system, there may be an aggravated over-reliance on private healthcare actors who have the means to support the digital healthcare infrastructure. This is likely to further widen the healthcare gap and have a disproportionally adverse impact on those already farthest removed from the healthcare system. The challenge will be to motivate and encourage key stakeholders, including patients, medical service providers, insurance companies and the government, to “pull as well as push the right kind of information from the system [83]”.

## 6. Recommendations for Policy and Legal Reforms

Before concentrating investment in large-scale efforts to digitize healthcare, the government and stakeholders in the healthcare space need to comprehensively evaluate existing legal and policy frameworks to ensure that digital health is approached under a ‘rights-based’ framework that prioritizes access to quality healthcare, user control over data, and the recognition of the rights to privacy and informed consent. The absence of uniform standards mandating the implementation of data protection and security [84] acts as a threat to the right to privacy, especially with respect to the NDHB’s propositions to link SPDI to private entities, such as insurers, pharmaceutical companies, and device manufacturers operating without safeguards. 

First, there should be proper regulation of personal health information, protecting patient safety and privacy while allowing for innovation. Data records for each individual need to be accurate, up-to-date, and structured in such a way that they can be assessed, modified, and deleted upon request. This requires individuals to have exclusive rights over data that has already been provided to service providers and key stakeholders. People using health data should be trained, and appropriate data privacy controls, such as user authentication and authorization whilst logging in, are essential to ensure informed consent, privacy, and capacity-building for the effective collection and processing of data. Informed consent must be facilitated through structured and culture-specific mechanisms for the use of patient data, and, wherever possible, data should be anonymized or aggregated. Ongoing monitoring processes and regular data audits should be mandated to ensure compliance with policies and procedures foregrounded within a rights-based approach to ensure the fundamental principles of autonomy, privacy, and self-determination.

Second, transparency, data confidentiality, and cyber security must be addressed in the guidelines concerning the health data of individuals to be used by medical software (third parties), and this must have a transparent data sharing agreement with the rights of patients being protected [45]. Scholars argue for the pressing need for a rights-based framework in the use of data and ownership of the data [85].

Third, the issue of liability must be addressed. Shoshana Zuboff terms this era as one of “surveillance capitalism”, which is a legitimate concern regarding the usage and processing of individual, community, and societal data [86]. Regulations for managing the handling of digital health technologies and big data not only foster user trust in digital health and thus the adoption of it but can also contribute to fair application and use of digital health.

Fourth, justice and other ethical concepts within digital health are under-discussed in the academic literature so far and overlooked in practice [87]. Especially from a public health point of view, ethical analyses to identify and remedy violations of justice, respect for autonomy, and other key values that form the basis for digital health initiatives, need to be adopted as a necessary and integral aspect of all further research [87]. It would be better, not just from the policy perspective but also regarding its application and functioning, to address issues of the digital divide and other infrastructural issues in India first.

Fifth, investment in digital healthcare interventions targeting marginalized persons that are culturally sensitive, respectful of traditional methods, and geared towards long-term capacity-building, support and robust monitoring and evaluation can have a long-lasting impact on the comparatively poorer health indicators of these communities [88]. This requires a unique articulation of social justice that can respond to the growing reliance on data technologies and the resultant rights concerns that they raise.

Finally, digital health is not just about the use of technologies for healthcare but is also contingent on the norms and culture of communities and people. Indigenous communities have been practicing herbal medicine and naturopathy historically but are being systematically deprived of these resources due to the corporatization of herbal resources, medicinal plants, and a focus on allopathic practices due to the exclusion of all other traditional medicinal practices. These communities are far more vulnerable to further discrimination based on the information that they may contribute when interacting with digital healthcare technologies. Further, the reliance of e-Health on internet connectivity and possession of basic devices to facilitate digitization is highly elitist in nature, as it will benefit only those (mostly in urban areas) with access to the requisite infrastructure. Therefore, NDHM’s presumably noble aims and objectives will be outweighed by its repercussions on marginalized individuals and communities if the needs of these people are not identified and adequately addressed in procedures and policy development.

When it comes to the digitization of health in India, the lack of a robust system coupled with the gaps in access to health that are the result of embedded and intersectional socio-economic inequality prove to be a significant challenge. Therefore, it is pertinent that any significant leap towards digitization of health must be preceded by a comprehensive regulatory framework that adequately addresses issues of informed consent, data protection, privacy, and accessibility. It must also adopt an intersectional rights-based approach towards developing the infrastructure that is necessary for such a leap to see the benefits translate on the ground.

## 7. Conclusions

The digitization of healthcare has the potential to increase accessibility to quality healthcare in India, which could greatly benefit marginalized individuals and communities. However, the legal and ethical challenges to implementing it must be addressed along with other structural challenges. Digitally aware healthcare workers are essential when conducting tests or administering procedures, including household health screenings, but the infrastructure of the country also needs to adapt to support digital health solutions. As discussed above, it is evident that digital healthcare succeeds in countries that invest well in the healthcare sector, such as Malaysia. The COVID-19 pandemic emphasized the importance of countries establishing adequate telemedicine practices through which doctors can contact their patients. However, in India, data protection regulations need to take care of the multiple challenges outlined above. Further, disparities in healthcare access and distrust in public facilities make the universal implementation of digital healthcare challenging.

In addition to legal challenges, there are ethical implications related to the digitization of healthcare in India. The primary ethical considerations concern the issues of informed consent, and these are critical concerns, particularly important for marginalized persons with low literacy rates, as well as communities that have historically been subject to medical exploitation. For the benefits of digital healthcare to reach those farthest removed from access to quality healthcare, there needs to be a comprehensive data protection and informed consent framework in place.

Therefore, it is imperative to implement a rights-based framework for health to prioritize informed consent, control user data, and promote the right to access healthcare being explicitly implemented in the context of the fundamental right to health as per constitutional interpretation. Scholars have pointed out the importance of ‘justice’ and ‘ethics’ as concepts that are currently underdiscussed in the academic literature so far and overlooked in practice, with reservations expressed on the NDHM’s disregard for fundamental infrastructural and socio-cultural barriers that the healthcare sector is plagued with today. Most importantly, there is a need for digital healthcare interventions to account for the intersection between caste, indigenous communities, and healthcare through the application of the five dimensions of access to healthcare, namely approachability, acceptability, availability and accommodation, affordability, and appropriateness.

## Figures and Tables

**Table 1 healthcare-11-00911-t001:** Indian policies affecting the digitization of healthcare.

Type of Regulation	Title of Regulation	Date Effective	Relevant Clauses
Law	The Drugs and Cosmetics Act (“D&C Act”)	10 April 1940	—
Law	Information Technology Act and Rules (IT Act)	9 June 2000	Section 2(w), Section 43A, Section 79
Regulations	The Clinical Establishments (Registration and Regulation) Act	9 August 2010	Section 38(1) and 38(2)
Law	The Information Technology Reasonable security practices and procedures and sensitive personal data or information Rules (“Data Protection Rules”)	1 April 2011	Rule 3, Rule 4(1), Rule 5(1), Rule 5(3), Rule 5(7), Rule 7
National Standards	The Information Technology (Intermediaries Guidelines) Rules (“Intermediary Guidelines”)	1 April 2011	Rule 3
Law	The Medical Devices Rules (“MDR”)	5 May 2017	—
Regulations	DNA Technology (Use and Application) Regulation Bill	8 July 2019	—
Regulations	Digital Personal Data Protection Bill (DPDP Bill)	18 November 2022	Clause 8

## Data Availability

The data supporting reported results of this article are available from the author (djain@jgu.edu.in).

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
