# Peer review of "Regulation of Digital Healthcare in India: Ethical and Legal Challenges"

_healthcare, 2023, doi:10.3390/healthcare11060911_

Round 1

Reviewer 1 Report

This is an important issue, and a timely paper. At present, the paper needs some additional work to clearly make its argument.

1) A definition of what you mean by health digitisation in the introduction is essential. Different scholars and policy makers use a range of definitions, so you need to make clear what is in scope, and what is out of scope, for your paper.

2) You need to reference your sources in the introduction. Currently there is no citation.

3) The introduction would be strengthened by grouping some of the ideas around important themes, eg access, data security and privacy concerns etc. They don’t all require subheadings, but at the moment the introduction lacks structure and tends to jump around from one idea to the next.

4) Do you have ethics approval for your interviews? If not- was it not required? You need to explain this

5)The article doesn’t currently make use of the interview data. Much of the information presented appears to have come from doing a desktop law and policy review. If you have good interview data, you need to use it- for example introduce the ‘theory’ (ie law and policy provisions); then present interviewee perspectives on those issues (using interview quotes is a really good approach here); then provide some analysis linking the interviewee views and the theory- do the interviews show that the law is working/not working? What challenges or benefits does that framework provide with respect to implementing digital health across India? 

Much of the descriptive legal and policy material presented in the results should really be presented in the introduction. The results for your paper should be where you present your interviewee responses to those laws and policies, and analyse their application to the issue (the ‘so what?’ Aspect of your paper).

6) Conclusion

What does your interview data tell you should be done? What are the main barriers to implementation, and benefits? This ideally should give the reader a summary of the main tasks that need to be done before effective implementation of digital health can occur- ie what laws and policies need changing, how much budget allocation does it require etc. You might conclude by saying that despite the requirements for change the benefits still exceed costs; or its value is unclear; or whatever conclusion your research leads you to draw. 

It is definitely worth persevering with this paper- all the background and literature is there, it just requires restructuring, and you need to get a lot more value from your interview data by working comments and quotes throughout your results section. 

Author Response

This is an important issue, and a timely paper. At present, the paper needs some additional work to clearly make its argument.

  • A definition of what you mean by health digitisation in the introduction is essential. Different scholars and policy makers use a range of definitions, so you need to make clear what is in scope, and what is out of scope, for your paper.

Author Response: This feedback has been incorporated and a section has been included in the beginning of the draft that details the definition of digital health to help contextualize the issues being discussed.

  • You need to reference your sources in the introduction. Currently there is no citation.

Author Response: All sources have been cited and the reference list has been updated.

  • The introduction would be strengthened by grouping some of the ideas around important themes, eg access, data security and privacy concerns etc. They don’t all require subheadings, but at the moment the introduction lacks structure and tends to jump around from one idea to the next.

Author Response: The draft has been restructured keeping in mind this feedback and the introduction has been revised to make the section more concise and focused with supplemental issues being addressed in the footnotes/end notes.

  • Do you have ethics approval for your interviews? If not- was it not required? You need to explain this

Author Response: Approval was taken for each of the interviews and the respondents were also informed of the purpose of the interviews giving them clarity as to what the material will be used for.

5)The article doesn’t currently make use of the interview data. Much of the information presented appears to have come from doing a desktop law and policy review. If you have good interview data, you need to use it- for example introduce the ‘theory’ (ie law and policy provisions); then present interviewee perspectives on those issues (using interview quotes is a really good approach here); then provide some analysis linking the interviewee views and the theory- do the interviews show that the law is working/not working? What challenges or benefits does that framework provide with respect to implementing digital health across India? Much of the descriptive legal and policy material presented in the results should really be presented in the introduction. The results for your paper should be where you present your interviewee responses to those laws and policies, and analyse their application to the issue (the ‘so what?’ Aspect of your paper).

Author Response:  Keeping in the mind the reviewer comments, the author has chosen to restructure the draft and remove the interview analysis all together. This decision was made taking note of the extensive critical analysis of the existing legal and policy frameworks that the draft can contribute towards in its attempt to covey the significant gaps and challenges that digitization of healthcare in India confronts. It is also in line with the publications previously published by the journal on this thematic issue and the author relied on two existing publications from China on the topic as a reference point for restructuring the draft.  Keeping the reviewer comment in mind, the author attempted to integrate the interview with theory but it became very cumbersome and incoherent, and very long. Upon reconsideration, the focus of the interviews was steering in multiple directions and required a very different conceptual framework to be able to do justice to all the interviewees and their concerns.  The author, therefore, focused on the reviewer suggestion to flesh out the question, What challenges or benefits does that framework provide with respect to implementing digital health across India? The paper now has a crisp introduction, a detail analysis of the legal and ethical challenges in Indian laws, moves beyond descriptive work and do a detail analysis of the data protection framework and its limitations, and critique of the law, the limitations of the current framework, challenges in implementation and policy recommendations within a rights-based approach. This is now one of the first pieces of comprehensive work on regulation digital health in India and reads much better without the interviews and is coherent.

6) Conclusion

What does your interview data tell you should be done? What are the main barriers to implementation, and benefits? This ideally should give the reader a summary of the main tasks that need to be done before effective implementation of digital health can occur- ie what laws and policies need changing, how much budget allocation does it require etc. You might conclude by saying that despite the requirements for change the benefits still exceed costs; or its value is unclear; or whatever conclusion your research leads you to draw. 

Author Response: This section has been revised in line with the updated structure which no longer uses the material from the interviews.

It is definitely worth persevering with this paper- all the background and literature is there, it just requires restructuring, and you need to get a lot more value from your interview data by working comments and quotes throughout your results section. 

Reviewer 2 Report

This paper focuses on "digitazion of healthcare" in India discussing different issues related to this process, also on the basis of a stakeholder consultation. There is a major problem that affects the quality of presentation: a clear definition of the scope of the research is missing. What does the author exactly mean with "Digital healthcare"?

Other  terms are introduced without specific definitions (health aggregator, m-health platforms, telemedicine) so that the section on findings and discussion is quite confusing.

Specific remarks:

- why the DNA technology Regulation bill is considered relevant? The relation with e-health is not clear

- It is not clear why lines 510-522 are under a subsection called: Key legal and policy etc...

- lines 649 - 659 are not in line with the topic of the sub-section (privacy)

- Section 5  is very poor.

- lines 110-112 are not clear 

Author Response

This paper focuses on “digitization of healthcare” in India discussing different issues related to this process, also on the basis oof a stakeholder consultation. There is a major problem that affects the quality of presentation” a clear definition and scope of research is missing. What does the author exactly mean by “digital healthcare”?

Author Response: This comment was very helpful in restructuring the paper. Upon re reading the paper, I agree with reviewer 2 that the paper lacked focus and become very incoherent in parts mostly due to the stakeholder consultations. Upon reconsideration, the focus of the interviews was steering in multiple directions and required a very different conceptual framework to be able to do justice to all the interviewees and their concerns.  The author, therefore, focused on the reviewer suggestion to flesh out the question, What challenges or benefits does that framework provide with respect to implementing digital health across India?  Keeping in the mind the reviewer comments, the author has chosen to restructure the draft and remove the interview analysis all together. This decision was made taking note of the extensive critical analysis of the existing legal and policy frameworks that the draft can contribute towards in its attempt to covey the significant gaps and challenges that digitization of healthcare in India confronts. It is also in line with the publications previously published by the journal on this thematic issue and the author relied on two existing publications from China on the topic as a reference point for restructuring the draft.  The draft is very coherent now.

A special section on defining digital healthcare has been included right after the introduction.

Other terms are introduced without specific definitions (health aggregator, m-health platforms, telemedicine) so that the section on findings and discussion is quite confusing.

Author Response: These have been fleshed out and defined in the restructuring process.

Specific remarks:

-Why the DNA technology regulation bill is considered relevant? The relations with e e health is not clear.

Author Response: The section on DNA section on page 7 and 8 has been fleshed out to achieve clarity and this section has been revised to discuss each regulation clearly.

It is not clear why lines 510-522 are under a subsection called: Key legal and policy etc... - lines 649 - 659 are not in line with the topic of the sub-section (privacy)

Author Response:  This section has been restructured and substantially revised now to achieve clarity including the section on privacy.

- Section 5 is very poor.

Author Response: This section has been restructured and substantially revised now to achieve clarity.

- lines 110-112 are not clear

Author Response: This section has been restructured and substantially revised now to achieve clarity.

Round 2

Reviewer 1 Report

The paper benefits greatly from the revisions, and provides a valuable insight into the legal and policy environment of health digitisation in one of the world's largest populations. 

Reviewer 2 Report

The manuscript has been sufficiently improved to warrant publication in Healthcare.